# ABC Transporters and CYP3A4 Mediate Drug Interactions between Enrofloxacin and Salinomycin Leading to Increased Risk of Drug Residues and Resistance

**DOI:** 10.3390/antibiotics12020403

**Published:** 2023-02-17

**Authors:** Min Chen, Yujuan Yang, Yupeng Ying, Jiamin Huang, Mengyuan Sun, Mian Hong, Haizhen Wang, Shuyu Xie, Dongmei Chen

**Affiliations:** 1National Reference Laboratory of Veterinary Drug Residues (HZAU), Wuhan 430070, China; 2MOA Laboratory for Risk Assessment of Quality and Safety of Livestock and Poultry Products, Huazhong Agricultural University, Wuhan 430070, China

**Keywords:** enrofloxacin, salinomycin, drug–drug interactions, CYP3A4, ABC transporter, steric-like effect

## Abstract

Enrofloxacin (ENR) is one of the most common drugs used in poultry production to treat bacterial diseases, and there is a high risk of drug interactions (DDIs) between polyether anticoccidial drugs added to poultry feed over time. This may affect the efficacy of antibiotics or lead to toxicity, posing a potential risk to the environment and food safety. This study aimed to investigate the DDI of ENR and salinomycin (SAL) in broilers and the mechanism of their DDI. We found that SAL increased the area under the curve and elimination half-life of ENR and ciprofloxacin (CIP) by 1.3 and 2.4 times, 1.2 and 2.5 times, respectively. Cytochrome 3A4 (CYP3A4), p-glycoprotein (P-gp) and breast cancer resistance protein (BCRP) were important factors for the DDI between ENR and SAL in broilers. ENR and SAL are substrates of CYP3A4, P-gp and BCRP in broilers; ENR and SAL inhibited the expression of CYP3A4 activity in a time- and concentration-dependent. Meanwhile, ENR downregulated the expression of P-gp and BCRP in a time- and concentration-dependent manner. A single oral administration of SAL inhibited CYP3A4, P-gp, and BCRP, but long-term mixed feeding upregulated the expression of CYP3A4, P-gp, and BCRP. Molecular docking revealed that ENR and SAL compete with each other for CYP3A4 to affect hepatic metabolism, and compete with ATP for P-gp and BCRP binding sites to inhibit efflux. ENR and SAL in broilers can lead to severe DDI. Drug residues and resistance following co-administration of ENR and SAL and other SAL-based drug-feed interactions warrant further study.

## 1. Introduction

Poultry has always remained the most commonly consumed meat around the world. According to the Food and Agriculture Organization of the United Nations (FAO), chicken consumption exceeded USD22.7 billion in 2016 and production reached 133.3 million tons in 2020 (FAO, 2018). The consumption of poultry meat has been increasing in recent years, but there are many problems in the poultry breeding process, such as *mycoplasma*, *salmonella*, *Escherichia coli* and other bacterial diseases, which have caused huge economic losses to the poultry breeding industry [1]. The global poultry industry is reported to lose millions of dollars annually to avian colibacillosis [2]. In addition to this, coccidiosis has long been one of the most expensive parasitic diseases in commercial poultry, with global economic losses from chicken coccidiosis from $3 billion in 1995 to over $13 billion in 2016 [3]. Intensive chicken raising is currently heavily dependent on antibiotics and anticoccidial drugs to control bacterial diseases and coccidiosis. 

Enrofloxacin (ENR) belongs to the quinolone class and is a broad-spectrum animal-specific antibiotic. ENR exhibits high antibacterial activity against *Mycoplasma*, *Salmonella*, *Campylobacter jejuni* and *Escherichia coli* by blocking DNA replication, and has always been one of the drugs of choice for the treatment of bacterial diseases in poultry [4]. Since coccidiosis can inhibit the growth of and cause a high mortality rate in chickens, in order to control coccidiosis in poultry, producers have added anticoccidial drugs to the feed to prevent coccidiosis [5]. According to the European Community (EC), about 45% of poultry feeds contain coccidiostatic drugs. Regulation (EU) No. 1831/2003 on Animal Nutritional Additives states that ionophores (monensin (MON), salinomycin (SAL), maduramycin (MAD), lassamycin (LAS)) anticoccidial drugs can be used as authorized compounds for poultry feed additives (EU 1831/2003). However, inappropriate, unreasonable, and prolonged use of these drugs or drug–drug interactions (DDIs) can lead to the accumulation of toxic and harmful residues in the meat and eggs of treated poultry. Studies have reported that the use of ENR in poultry farming induces quinolone resistance in *Campylobacter jejuni*, which is then transferred to humans through poultry exposure and leads to treatment failure in human campylobacteriosis [6]. In addition to drug resistance reported globally, polyether drugs may also pose a threat to consumers’ health through tissue residues. Long-term exposure to low-level polyether-based drugs may lead to chronic toxicity, carcinogenic effects, and other unclear consequences [7]. However, to date, despite the polyethers and ENRs being the drugs that often coexist for a long time in poultry farming, there is a lack of information on safety assurances when shared, such as potential DDI.

Among the mechanisms of DDI production, the hepatic and intestinal cytochrome P450 system and transporter system are the most common. According to previous reports, ENR has been reported to be a cytochrome P450 enzyme inhibitor that has the potential for interaction with other drugs [8]. Zhang reported that ENR could reduce the expression of cytochrome P1A (CYP1A) and 3A (CYP3A) proteins in broilers and then increase the elimination half-life of caffeine and dapsone [9], which indicates that ENR may be at risk of drug interactions with CYP3A substrates. MON, SAL and MAD could induce CYP3A protein expression in broilers. In addition, ENR is also a substrate for pig cytochrome P3A4 (CYP3A4) [8], and ENR and SAL are substrates for human CYP3A4 [10]. Although no studies reported that ENR and SAL were substrates of CYP3A4 in broiler chickens, CYP3A4 sequences from different species had high sequence homology, meaning that ENR and SAL are also highly likely to cause CYP3A4-based DDI in chickens. Meanwhile, ABC (ATP-binding cassette) transporters (P-glycoprotein (P-gp, ABCB1) and breast cancer resistance protein (BCRP, ABCG2)) localize to polarized epithelia (eg, gut, liver, kidney, apical membranes of the placenta, mammary gland, and blood–brain barrier), which can modulate the pharmacokinetics of drugs, leading to the development of DDI of different compounds [11,12]. SAL is a potent transport substrate for mouse and human P-gp [7] and is a major determinant of organic anion pharmacokinetics and toxicity. Changes in P-gp activity may directly affect the effective exposure of SAL and substrate drugs. In 2011, Kim et al. demonstrated that SAL can effectively inhibit human P-gp, even more potently than the well-known P-gp inhibitor verapamil (Ver) [13]. While ENR has also been identified as a chicken and porcine P-gp substrate and an in vitro substrate of cow, ovine and chicken ABCG2 [14,15], its pharmacokinetics in animals correlate with the expression levels of P-gp and BCRP. The above studies show that polyethers and ENRs are likely to interact with the transporters ABCB1 and ABCG2, thereby affecting the pharmacokinetics of the drug.

As chemical-sensing transcriptional factors, xenogenic receptors (such as nuclear receptors pregnane X receptor (PXR), constitutive androstane receptor (CAR), and non-nuclear receptor aryl hydrocarbon receptor (AHR)) play a crucial role in the transcriptional regulation of drug-metabolizing enzymes and drug transporters. These receptors can be activated by exogenous substances to regulate the expression of drug-metabolizing enzymes and transporters, thereby influencing drug metabolism and disposal. The activation of PXR induces the expression of phase I Cyp3a, Cyp2b6, Cyp2c9 and Cyp2c19; phase II glutathione *S*-transferase, UDP-glucuronic transferase, sulfonyl transferase; and phase III multidrug resistant protein, organic anion transport polypeptide 2, P-gp and BCRP [16,17]. PXR and CAR also play a crucial role in the induction of BCRP in rodents and humans [18,19]. The phase I metabolic enzymes, Cyp1a and Cyp1b, are the main target genes of AHR [20]. The xenogenic receptor is an important mediator in the process of drug metabolism and transport, and is one of the important reasons for mediating CYP450 enzyme and drug transporter DDI. 

However, to date, there has been no study on whether DDI is caused by interaction between ENR and polyethers, and little is known about the possible mechanisms of DDI between the two. Thus, this study aimed to investigate the effects of ENR and SAL on the expression levels of CYP450 and transporter in chickens, both in vitro and in vivo, to explore the pharmacokinetic interaction of ENR and feed additive SAL and the possible impact of transporters on the clinical efficacy of ENR and SAL. Combined with molecular docking technology, the binding conformation between drugs and metabolic enzymes and transporters was analyzed from the perspective of protein structure.

We emphasize that, in this study, we propose for the first time the risk of DDIs between SAL and ENR based on metabolic enzymes and transporters, and further clarified the molecular mechanism of DDIs and the “steric effect” of ENR and SAL. This work will guide the poultry industry in the rational use of ENR and feed additives SAL. The can help reduce to the toxicity of SAL to broiler chickens and humans, and also reduce the residue of ENR excess, thereby limiting the spread of antibiotic resistance and ensuring food safety.

## 2. Materials and Methods

### 2.1. Animal and Cell Culture

A total of 30 Arbor Acres (AA) broilers, 35 days old, 1.0–1.5 kg, were purchased from the Experiment Animal Center of Huazhong agricultural University. The broilers were raised and managed in the laboratory animal room of the National Reference Laboratory for Residues of Veterinary Drugs (HZAU). All chickens had free access to water for 7 days in an environment with relative humidity (45–65%) and temperature (18–25 °C).

Human colorectal adenocarcinoma cells (Caco-2 cells) were purchased from Procell (Wuhan, China) and grown using DMEM (Gibco, Shanghai, China) containing 10% fetal bovine serum (FBS, PAN, Adenbach, Germany) and 1% 100 U/mL penicillin/streptomycin (Gibco, Shanghai, China), and cultured in a 37 °C constant temperature incubator containing 5% CO_2_. The number of passages when the cells were purchased back was 5, and the passage of Caco-2 cells at the time of stimulation by ENR and SAL amounted to approximately 12–14 passages. Cells could adhere to the wall about 6–7 h after passage, and grow to 80% by 24–36 h.

### 2.2. HPLC or LC-MS/MS Analysis

The chromatographic system used in samples analysis consisted of a Waters 2475 separation module and a Waters 2998 UV/ fluorescence detector. A Shimadzu high performance liquid chromatography (HPLC) system and API 5000 mass spectrograph were used to liquid chromatography with tandem mass spectrometry (LC/MS/MS) analysis. The separate system of LC-MS/MS and HPLC were equipped with a Thermo Hypersil GOLD C18 (2.1 × 150 mm, 5 μm) and C18 column (SB-Aq, 250 × 4.6 mm, i.d., 5 μm, Agilent, Santa Clara, CA, USA), separately.

ENR and CIP (Dr. Ehrenstorfer Gmbh, Augsburg, Germany): which were performed with an excitation wavelength of 280 nm and an emission wavelength at 450 nm. Mobile phase compositions were 0.5 mol/L H_3_PO_4_-triethylamine (pH = 2.4, phase A) and acetonitrile (phase B) (82:18, *v/v*) [1,18].

SAL (Shandong Lukang, Jining, China): The LC-MS/MS assay method for SAL were introduced in the previous research [21]. The separate system of HPLC used 0.1% formic acid solution (phase A) and methanol (phase B) as mobile phase. The gradient condition: 0 min-40% phase B, 2 min-40% phase B, 3 min-100% phase B, 12 min-100% phase B, 13 min-40% phase B, 15 min-40% phase B. The flow rate is 0.2 mL/min, and the quantitative analysis of ion pair *m/z* is 773.5/431.3. Positive ion scanning mode was used; DP, CE, and CXP are 59.1 V, 65.64 V, and 12 V, respectively.

Coumarin (CHO, Sigma-Aldrich, Saint Louis, MO, USA): The sample was detected with an excitation wavelength of 326 nm and an emission wavelength at 460 nm. Mobile phase compositions were 0.1% formic acid water (phase A) and acetonitrile (phase B). The gradient condition: 0 min-20% phase B, 5 min-20% phase B, 20 min-30% phase B.

Testosterone (TS, Sigma-Aldrich, USA): The wavelength was 254 nm. Mobile phase compositions were water (phase A) and acetonitrile (phase B) (50:50, *v/v*).

Dextromethorphan (DM, Sigma-Aldrich, Saint Louis, MO, USA), diclofenac (DCF, Sigma-Aldrich, Saint Louis, MO, USA), mephenytoin (MP, Sigma-Aldrich, Saint Louis, MO, USA) and chlorzoxazone (CLZ, Sigma-Aldrich, Saint Louis, MO, USA): The LC-MS/MS used 0.1% formic acid solution (phase A) and methanol (phase B) as mobile phase. The gradient condition: 0 min-5% B, 2 min-5% B, 3 min-50% B, 5 min-60% B, 7 min-80% B, 10 min-80% B, 11 min-5% B, 15 min-5% B.

### 2.3. Sample Preparation

Plasma sample: Add 0.8 mL of acetonitrile to 0.2 mL of plasma to precipitate proteins. After vortexing the mixture for 5 min, centrifuge at 12,000 rpm for 10 min. Transfer the supernatant to a new 10 mL tube. The supernatant was blown dry with nitrogen at 40 °C, and the pellet was redissolved with 1 mL of methanol. The methanol reconstituted solution was filtered through a 0.2 μm filter.

Probe drugs: a volume of 0.2 mL cold acetonitrile were added into 0.05 mL microsome incubation solution, then vortexed for 2 min to extract target compounds. The mixture was centrifuged at 12,000 rpm for 15 min. The supernatant was filtered through a 0.2 μm filter.

### 2.4. Effects of Polyethers and ENR on CYP450 Enzymes Activity

Add 0.1, 0.5, 1.0, 1.5 and 2.0 mg/mL broiler liver microsomes to 0.05 mmol/L Tris-HCl buffer and 1 mmol/L NADPH system, and pre-incubate for 5 min. SAL and ENR were, pre-incubated at 41 °C for 30 min before addition substrates, respectively. Additionally, then DCF (0.625, 1.25 and 2.5 μg/mL), MP (0.0125, 0.025, 0.05 μg/mL), DM (0.0625, 0.125 and 0.25 μg/mL), and CLZ (0.625, 1.25 and 2.5 μg/mL) were added into and incubated at 41 °C for 0.5, 12, 4, 8 and 24 h. Coumarin (5, 10, 15 and 20 μmol/L) and TS (0.25, 0.5 and 1 mg/mL) were added into and incubated at 41 °C for 5, 15, 30 and 60 min. The total reaction system was 500 µL, SAL was prepared with methanol, and the final volume fraction of methanol in the reaction system was less than 1%.

### 2.5. Type of Action of SAL and ENR on CYP450 Enzymes

Final concentrations of 50, 100 and 500 μmol/L SAL and ENR were pre-incubated at 41 °C for 0, 3, 5, 15, 30 min before adding probe substrate. Except for testosterone, which was incubated for 30 min, all other probe drugs were incubated for 4 h. 

### 2.6. Screening of CYP450 Enzymes That Metabolize SAL and ENR in Broilers

The known positive inhibitors of CYP450 were adopted to determine the main metabolic enzymes of SAL and ENR. Briefly, alpha-naphthoflavone (α-NIF, inhibitor of CYP1A2), sulfafenpyrazole (SUl, CYP2D6), sodium diethyldithiocarbamate (DIE, inhibitor of CYP2E1), omeprazole (OMP, inhibitor of CYP2C45) and ketoconazole (KCZ, inhibitor of CYP3A4) were incubated with 50 μmol/L SAL or the ENR-NADPH system, respectively. The system consisted of 0.5 mg/mL broiler liver microsomes, 0.05 mmol/L Tris-HCl buffer, and 1 mmol/L NADPH. Before the addition of SAL or ENR, inhibitors were pre-incubated at 41 °C for 30 min, and then incubated with SAL or ENR for 4 h.

### 2.7. Effects of ENR and SAL on Each Other’s Metabolism In Vitro

Final concentrations of 50 μmol/L SAL and ENR were incubated with NADPH system consisting of 0.5 mg/mL broiler liver microsomes, 0.05 mmol/L Tris-HCl buffer, and 1 mmol/L NADPH. Final concentrations of 50 μmol/L SAL or 50 μmol/L ENR were pre-incubated at 41 °C for 30 min before the addition of another drug (ENR or SAL), and were then incubated for up to 4 h.

### 2.8. Pharmacokinetic Studies of ENR and SAL in Broilers

A total of 30 AA broilers were divided into 6 groups including a control group (gavage with equal volume of normal saline), single ENR group (gavage with 7.5 mg/kg), single SAL group (gavage with 6.0 mg/kg), ENR (7.5 mg/kg) + SAL (6.0 mg/kg) co-gavage group (ENR + SAL), ENR pre-treatment for 5 days (75 mg/L mixed drink) + SAL (gavage with 6.0 mg/kg) group (ENR (5d) + SAL), and a SAL pre-treatment for 5 days (60 mg/kg mixed feeding) + ENR (gavage with 7.5 mg/kg) group (SAL (5d) + ENR). Volume 2 mL blood samples were collected at 0.5, 1, 2, 4, 8, 12, 24, and 48 h after administration, placed in a centrifuge tube with sodium heparin, and centrifuged at 4 °C, 4000 rpm for 10 min to separate plasma. PK parameters were calculated with WinNonlin software using non-compartmental settings (version 6.4; Pharsight Corporation, Mountain View, CA, USA).

### 2.9. RT-PCR and Western Blot

The relative expressions of CYP3A4, P-gp and BCRP in tissue of broiler and Caco-2 were evaluated via quantitative real-time PCR (RT–qPCR, Bio-Rad, Hercules, CA, USA). When culturing Caco-2 cells to 70% of the plate, incubate with the ENR(5 μM and 20 μM) or SAL (0.5 μM and 2 μM) for 12 and 24 h, respectively. Relevant cells were treated as indicated and total RNA was extracted from cells and tissue using RNA isolater Total RNA Extraction Reagent (R401-01, Vazyme Biotech Co., Ltd., Nanjing, China) following the manufacturer’s protocols, then reverse-transcribed into cDNA using HiScript II Q Select RT SuperMix for Qpcr (R233-01, Vazyme Biotech Co., Ltd.). The cDNA was determined by quantitative RT-PCR using 2 × AceQ qPCR SYBR Green Master Mix (Q141-02, Vazyme Biotech Co., Ltd.) with the primers (listed in Appendix A). All data were analyzed using the 2−ΔΔCt method [22]. 

Caco-2 cells were stimulated by ENR (5 μM and 20 μM) and SAL (0.5 μM and 2 μM) for 12 and 24 h, as indicated, and the drug-treated tissues were sheared and ground. These were then collected in a RIPA lysis buffer (P0013B, Beyotime, Shanghai, China) and incubated on a rocker with ice for 30 min. Cell and tissue lysates were centrifuged at 12,000× *g*/4 °C for 15 min and lysates were boiled at 100 °C for 5 min in 5× SDS Loading Buffer and resolved by SDS-PAGE. Proteins were transferred into a polyvinylidene fluoride (PVDF) membrane (Bio-Rad) and then incubated with the CYP3A4 Rabbit pAb (Cat#A2544, ABclonal, Wuhan, China), ABCG2 Rabbit pAb (Cat#A5661, ABclonal, Wuhan, China), P-gp Rabbit pAb (Cat#A11747, ABclonal, Wuhan, China) over night, respectively. They were then incubated with the HRP Goat Anti-Rabbit IgG (Cat#AS014, ABclonal, Wuhan, China) for 1.5 h. Immobilon Western Chemiluminescent HRP Substrate (Millipore) was used for protein detection. The immunoreactive bands were visualized on a gel imaging system (Tanon 5200 Multi, Tanon Technology Co., Ltd., Shanghai, China), and the intensities of the immunoreactive bands were quantified using Quantity One. 

### 2.10. Homology Modeling of CYP3A4 and P-gp and Molecular Docking with ENR and SAL

The structures of SAL and ENR were downloaded from NCBI (https://www.ncbi.nlm.nih.gov/, accessed on 24 June 2022) and MOPAC energy minimization was performed using Avogadro. Based on the amino acid sequences of gallus CYP3A4, P-gp and BCRP on NCBI, and the crystal structures of human CYP3A4 (PDB ID: 6MA7) [23], P-gp (PDB ID: 6C0V) [24] and BCRP (PDB ID: 6ETI) [25], the broiler CYP3A4, P-gp and BCRP were homologous modeled by Swiss-Model (https://swissmodel.expasy.org/, accessed on 25 June 2022). Modelling results were submitted to UCLA-DOE LAB-Laboratory Services (http://services.mbi.ucla.edu/, accessed on 25 June 2022) for evaluation. 

Active pockets and hydrogen bonding interactions between ENR and SAL and broiler CYP3A4, P-gp and BCRP were analyzed using SYBYL-X 2.0. The hydrogen bonding force between proteins and compounds was determined by generating a 2D interaction diagrams of proteins and ENR or SAL by Ligplot (LIGPLOT) home page (ebi.ac.uk).

### 2.11. Statistical Analysis

SPSS software (version 20, IBM, New York, NY, USA) was used for statistical analysis of experimental data. Two groups of data were analyzed for significance by *paired t* test (paired samples *t* test), and multiple groups of data were analyzed by one-way analysis of variance (One-Way ANOVA). Results are expressed as Mean ± SD, * means *p* < 0.05, ** means *p* < 0.01, *** means *p* < 0.005, **** means *p* < 0.001.

## 3. Results

### 3.1. ENR and SAL Cause Each Other’s Metabolism to Slow down In Vitro Based on CYP3A4

CYP450 enzymes are one of the most important mechanisms leading to DDI. Given that ENR is reported to be a substrate of human and porcine CYP3A4, we first verified that CYP3A4 is the main metabolic enzyme that metabolizes ENR in broilers by an enzyme-specific inhibitor method. The results are shown in Figure 1A. Compared with the ENR metabolite group, the ENR metabolic rate decreased by 32% after the addition of the CYP3A4-specific inhibitor KCZ, and soon displayed the lowest metabolic rate among all inhibitor groups.

Then, to explore whether polyether drugs and ENR may have potential drug interactions based on CYP450, we tested the effects of polyether drugs and ENR on CYP450 enzymatic activity in vitro using a probe substrate assay. The LC-MS/MS methods of the probe substrate method can be found in Appendix A in detail. Additionally, the methodological results for probe substrates are given in the Appendix A. Figure 1B–G were the inhibition rates of ENR and polyether drugs on CYP450 enzymes, respectively. Except for SAL and LAS, which have inductive effects on CYP2A6 and CYP2E1, respectively, all others showed inhibitory effects. In clinical practice, since inhibition of metabolic enzymes by DDI is usually more of a concern, we placed the focus on the inhibition of CYP450 enzymes. However, other than SAL, which has a strong inhibitory effect on CYP3A4 and can inhibit TS metabolism by 52%, the inhibition rate of other compounds on CYP450 enzymes is less than 30%. These results indicated that SAL may inhibit ENR metabolism by inhibiting CYP3A4. At the same time, we also found that ENR also has a 32% ± 3.1 inhibition rate on CYP3A4. Therefore, we continued to explore the CYP450 enzymes of SAL in broilers through in vitro inhibitor experiments. As shown in Figure 1H, KCZ significantly inhibited SAL metabolism by about 36%, confirming that SAL is a substrate of CYP3A4 in broilers. The above results demonstrated that ENR and SAL may have DDI based on CYP3A4. In addition, we also examined the time-dependent and concentration-dependent effects of ENR and SAL on CYP3A4. The results are exhibited in Figure 2. The inhibitory effects of ENR (Figure 2A) and SAL (Figure 2C) on CYP3A4 were time-dependent, and the longer the pre-incubation time, the stronger their inhibitory effects. However, ENR induced CYP3A4 at 0 and 3 min of pre-incubation. In addition, ENR (Figure 2B) and SAL (Figure 2D) exhibited concentration-dependent inhibitory effects on CYP3A4 at concentrations of 50, 100 and 500 μmol/L. As expected, ENR or SAL pretreatment significantly inhibited the metabolism of the other compounds studied, with ENR and SAL metabolism inhibited by 24.92% and 14.75%, respectively (Figure 2E,F). These results suggest that ENR and SAL are substrates of CYP3A4 in broilers and are also able to inhibit CYP3A4 enzymatic activity, so when ENR and SAL are used together, they can cause the metabolic inhibition of each other based on CYP3A4, leading to DDI. 

### 3.2. Pharmacokinetics Assessment after Co-Administration

#### 3.2.1. HPLC Methods of ENR and CIP

We used a standard curve to determine the concentrations of ENR and CIP. The linear range of ENR and CIP was 0.02–5.0 μg/mL, and the correlation coefficients (r) of ENR and CIP were 0.9998 and 0.9997, respectively. The limits of detection (LOD) were 0.01 μg/mL and 0.02 μg/mL, respectively. The recoveries of ENR and CIP at different concentrations in plasma were 95.7–107.0% and 96.2–106.9%, respectively, and the precision relative standard deviation (RSD) was less than 7.0%. See Appendix A for details.

#### 3.2.2. LC-MS/MS Methods of SAL

The standard curve of SAL was from 0.2 to 50 μg/L, and the correlation coefficient (r) for SAL standard curve was 1. Both LOD and limit of quantification (LOQ) values were 0.2 μg/L. The recovery rates of three different added concentrations in plasma for ENR and CIP were 96.19–105.05%, respectively, and the RSD of precision was less than 15%. See Appendix A for details.

#### 3.2.3. Effects of SAL on the Absorption and Elimination of ENR and CIP in Broilers

In order to systematically study the interaction between ENR and SAL, the in vivo pharmacokinetics of ENR and SAL in broilers were investigated. Plasma concentrations of ENR (Figure 3A) and the major metabolite CIP (Figure 3B) were significantly increased in the ENR + SAL group and the SAL (5d) + ENR group compared with the group of ENR alone group. Compared with the control group, the area under the curve values (AUC_0-∞_) of ENR in ENR + SAL group and SAL (5d) + ENR group were 1.29 times and 1.19 times, respectively (Appendix A). The maximum plasma concentration (C_max_) and AUC_0-∞_ values of CIP were 1.25 times and 2.75 times in the ENR + SAL group, respectively, and 1.38 and 2.34 times in the SAL (5d) + ENR group, respectively (Appendix A). At the same time, the pharmacokinetic parameters of CIP, such as terminal elimination rate (Ke), elimination half-life (T_1/2_) and mean residence time (MRT_0-∞_), related to drug elimination were significantly changed. Compared with the control group, Ke values in ENR + SAL group and SAL (5d) + ENR were significantly decreased by 2.67 times and 2.29 times, respectively. The T_1/2_ and MRT_0-∞_ values of the ENR + SAL group were increased by 2.47 times and 1.87 times, respectively; the T_1/2_ and MRT_0-∞_ of SAL (5d) + ENR group were increased by 2.21 times and 2.09 times, respectively. These results infer that the effect of SAL on ENR occurs in the gastrointestinal tract and liver, since SAL not only improves the absorption efficiency of ENR and CIP, but also slows down the elimination of CIP.

#### 3.2.4. Effects of ENR on the Absorption and Elimination of SAL in Broilers

The pharmacokinetic curves and pharmacokinetic parameters of SAL in broilers after different treatment groups are shown in Figure 3 and Appendix A, respectively. Compared with the SAL alone group, the AUC_0-∞_ and C_max_ of the ENR + SAL group were decreased by 42% and 33%, respectively; the AUC_0-∞_ and C_max_ of the ENR (5d) + SAL group were increased by 1.87 times and 1.64 times, respectively. The MRT_0-∞_ of the ENR + SAL group and the ENR (5d) + SAL group increased by 1.15 times and 2.41 times, respectively. In contrast, the plasma concentration of SAL decreased during in ENR + SAL group, and the combination of ENR for 5 consecutive days not only enhanced SAL absorption, but also decreased SAL elimination.

### 3.3. Effects of ENR and SAL on mRNA and Protein Expression of Hepatic CYP3A4 

To further explore how ENR and SAL affect each other’s pharmacokinetics, we here determined the protein and mRNA levels of CYP3A4 in the small intestine and liver. Figure 4 shows the results. Compared with the physiological saline group, ENR and SAL significantly induced and inhibited the protein expression of CYP3A4, respectively (Figure 4A,B). Relative to the single ENR group, ENR + SAL and SAL (5d) + ENR significantly inhibited the mRNA and protein expression of CYP3A4, which was consistent with the in vitro activity study, and the inhibitory effect of SAL on CYP3A4 was time- and concentration-dependent (Figure 4C). Contrasted with the single SAL group, overexpression of CYP3A4 gene was observed in the ENR + SAL group, and the ENR (5d) + SAL group significantly inhibited the mRNA and protein levels of CYP3A4 (Figure 4D).

### 3.4. Effects of ENR and SAL on mRNA and Protein Expression of P-gp and BCRP in Liver and Small Intestine 

Pharmacokinetic studies of ENR and SAL have shown that ENR or SAL can affect the absorption and elimination of SAL or ENR in vivo. Therefore, this work performed fluorescent quantification and immunoblotting for the expression of ABC transporters (P-gp and BCRP) in the liver and small intestine. The gene and protein expression levels of P-gp and BCRP in the liver and small intestine are shown in Figure 5 and Figure 6. The results showed that ENR and SAL were able to suppress P-gp gene expression levels in the liver and small intestine compared with the saline group (Figure 5A,D). P-gp levels in the small intestine were significantly suppressed in the ENR + SAL group compared with the ENR or SAL groups. Contrary to what was seen in the previous results, ENR (5d) + SAL group and SAL (5d) + ENR group significantly induced changes in the gene expression levels of P-gp in liver and small intestine. 

The effects of the ENR + SAL group and SAL (5d) + ENR group on BCRP gene levels were consistent with the results on P-gp (Figure 6B,E). Compared with the single ENR group, ENR + SAL significantly inhibited the expression of BCRP, but SAL (5d) + ENR group significantly induced the expression level of BCRP. Therefore, ENR showed a different effect on P-gp, and ENR appeared to inhibit the mRNA expression of BCRP in a concentration- and time-dependent manner. Compared with the single SAL group, the inhibition rates of BCRP expression in the ENR + SAL group and ENR (5d) + SAL were 23.02% and 74.94% (liver) and 65.49% and 76.78%, respectively (Figure 6C,F). 

The above results indicate that the gene expression of P-gp and BCRP in the liver and small intestine can be significantly or extremely significantly inhibited under the condition of low concentration SAL or SAL treatment for a short time. Conversely, high concentrations of SAL or prolonged presence of SAL may increase the mRNA expression levels of P-gp and BCRP in liver and small intestine. For ENR, the gene levels of P-gp and BCRP were decreased in a concentration- or time-dependent manner. Therefore, ENR and SAL may affect the absorption and transport of SAL and ENR in broilers by inhibiting P-gp and BCRP.

### 3.5. Effect of ENR and SAL on the Expression of mRNA and Protein in Caco-2 Cells

#### 3.5.1. Viability Assay of ENR and SAL on Caco-2 Cells

For a more intuitive understanding of the effects of ENR and SAL on P-gp and BCPR. We intend to perform validation on Caco-2 cells. Caco-2 cells were exposed to different doses of ENR (0, 5, 10, 20, 50, 100, 200 and 500 μM) and SAL (0, 0.05, 0.5, 5, 10, 25, 50 and 100 μmol/L) for 24 h and analyzed for cell viability using the CCK8 assay (Appendix A). The average viability of Caco-2 cells was over 90% after 5–20 μmol/L ENR treatment for 24 h, and the average Caco-2 cell viability was over 75% after 0.05–5 μmol/L SAL treatment for 24 h. Therefore, ENR at 5 and 20 μmol/L and SAL at 0.5 and 2 μmol/L were selected for follow-up studies. 

#### 3.5.2. Viability Assay of ENR and SAL on Caco-2 Cells

We treated Caco-2 cells with different concentrations of ENR (5 and 20 μmol/L) and SAL (0.5 and 2 μmol/L) for 12 and 24 h, respectively, and measured the gene (Figure 7A,B) and protein (Figure 7C,D) expression levels of P-gp and BCRP. The results of qPCR and WB showed that the gene and protein expressions of P-gp were significantly decreased after ENR and SAL treatment at the concentrations in this experiment for 12 or 24 h. However, this inhibition decreased with the increase of time and concentration. This phenomenon can also be seen in the expression level of BCRP pretreated with SAL. The difference was that ENR decreased the expression of BCRP in a concentration- and time-dependent manner. Combining the previous expression levels of P-gp and BCRP in liver and small intestine, we can think that low-concentration of SAL treatment inhibits the expression of P-gp and BCRP. However, ENR could induce the expression of P-gp and BCRP in a time- and concentration-dependent manner, as evidenced by reduced inhibition at low concentrations. ENR induced P-gp expression in a time- and concentration-dependent manner, and low concentrations exhibited P-gp inhibition. ENR induced time- and concentration-dependent reductions on BCRP expression.

### 3.6. Computer-Aided Molecular Docking Assays

#### 3.6.1. ENR Had a “Steric-like Effect” on SAL 

The studies in vitro have shown that ENR and SAL are substrates of CYP3A4 in broilers, so ENR and SAL are likely to compete with CYP3A4 and affect each other’s metabolism. Therefore, we explored the interaction between ENR, SAL and CYP3A4. Before docking CYP3A4 with drug molecules, we performed homology modeling on chicken CYP3A4, and the CYP3A4 (PDB ID: 6MA7) homology modeling score results are shown in Appendix A. The binding conformations of ENR and SAL to CYP3A4 are shown in Figure 8. The docking scores and hydrogen bond interactions of ENR and SAL with CYP3A4 (PDB ID: 6MA7) were predicted by molecular docking simulations, as shown in Appendix A. The results showed that SAL and ENR have a certain affinity with CYP3A4 (PDB ID: 6MA7). Moreover, there is a shared amino acid residue binding site (A/Glu 381) between ENR and SAL. These results indicate that ENR or SAL are likely to compete for the binding site of SAL or ENR and CYP3A4, forming a steric-like effect and inhibiting the metabolism of both ENR and SAL.

#### 3.6.2. ENR and SAL Had a “Steric-like Effect” on ATP for NBD Sites Causing the Inhibition P-gp and BCRP

The protein structures of chicken P-gp and BCRP were constructed by homology modeling of human ABCB1 (PDB ID: 6C0V) and human ABCG2 (PDB ID: 6ETI), and the results of homology modeling are shown in Appendix A, respectively. As shown by the docking results of ENR, SAL and ATP with chicken P-gp (Figure 9 and Appendix A) and BCRP (Figure 10 and Appendix A), ENR, SAL and ATP have good affinities for P-gp. Between ATP and P-gp, there are eight hydrogen bonds consisting of five amino acid residues, and the binding fraction is 9.1007. There are four and five intermolecular hydrogen bonds between ENR and SAL and P-gp, respectively. ENR, SAL and ATP all share the same amino acid residue binding sites, such as A/LYS181, A/GLU243 and A/LYS826 between ENR and ATP, and A/LYS181 between SAL and ATP. Likewise, ENR, SAL, and ATP share the same binding sites with BCRP, such as A/ASP237 (ENR and ATP) and A/LYS124 (SAL and ATP). The common amino acid sites of ENR, SAL and ATP on P-gp and BCRP suggest that ENR and SAL may compete with ATP for NBD sites on P-gp and BCRP, resulting in the inhibition of P-gp and BCRP drug efflux.

## 4. Discussion

As one of the most widely used drugs in the treatment of poultry bacterial diseases, ENR has an important position. Additionally, polyether anticoccidial drugs are approved as feed additives to prevent poultry coccidian. Despite the trend towards less extensive use of chemotherapy in poultry production, ENR and anticoccidial products are still used routinely [26] and are considered essential for broiler production in many parts of the world [27]. To date, little is known about the potential for drug–drug interactions of ENRs and polyethers in broilers. Since dysregulation of the CYP450 enzyme and transporter plays a critical role in altering the pharmacokinetics and pharmacodynamics of conventional drugs, DDI studies based on these two are critical.

Overviewing the CYP450 enzymes that metabolize ENR in broilers by chemical inhibitor method first, the results of the study found that CYP3A4 is the main metabolic enzyme that metabolizes ENR in broilers, and ketoconazole (CYP3A4 specific inhibitor) inhibits its metabolism by 32% (Figure 1A). This result is consistent with the metabolism of ENR in pigs and humans [8,10]. Then, we studied the effects of ENR and polyether drugs on CYP450 enzymes by the probe substrate method. We found that ENR and SAL could inhibit the CYP3A4 activity by 32% and 52% (Figure 1G), respectively, and the inhibition was time- and concentration-dependent (Figure 2). However, it is interesting that the activity of CYP3A4 is enhanced after low-concentration ENR and short-term incubation of ENR (Figure 2A,B). This phenomenon has also been reported before [28,29]. These findings indicate that ENR and SAL are highly likely to undergo DDI based on CYP3A4. Based on this idea, we continued to explore the effects of ENR and SAL on each other’s metabolism in an in vitro chicken liver microsomal system. Results: As expected, ENR or SAL pretreatment could significantly inhibit the metabolism of the other compound studied, and the metabolism of ENR and SAL were inhibited by 24.92% and 14.75%, respectively (Figure 2E,F). These results suggest that ENR and SAL are substrates of CYP3A4 in broilers, while simultaneously inhibiting CYP3A4. Therefore, when ENR and SAL are used together, metabolism will slow down due to the inhibition of CYP3A4, thereby increasing the plasma drug concentration in vivo.

We have shown in vitro that ENR and SAL cause each other’s metabolic rates to decrease. However, due to the optimized concentration of ENR and SAL in the incubation system in vitro and many influencing factors in vivo, the in vitro metabolic effects do not indicate the presence of DDI in vivo for ENR and SAL. Therefore, the effect of different exposure time of ENR or SAL on the pharmacokinetics of SAL or ENR was investigated in this work. The results showed that the pharmacokinetic parameters of SAL or ENR changed significantly after exposure to ENR or SAL. Plasma AUC_0-∞_ of ENR was significantly increased after SAL exposure (Appendix A). This result is consistent with the pharmacokinetic results of CYP3A4 inhibition by Liu et al. [30]. This suggests that SAL treatment reduces ENR metabolism, leading to an increase in its plasma concentration. In addition, we found that the AUC_0-∞_ of CIP was significantly increased, and its elimination-related pharmacokinetic parameters, such as MRT_0-∞_ and T_1/2_, were significantly increased, and Ke was decreased significantly (Appendix A). Compared to the ENR + SAL group, the plasma concentrations of ENR and CIP were slightly decreased in the SAL (5d) + ENR group. Considering the time- and concentration-dependent inhibition of CYP3A4 by SAL, we speculate that the above phenomenon may also be related to transporters in the gut and liver. ENR has also been previously reported to be a substrate of P-gp and BCRP in sheep and cows [14,15,31], and SAL has also been shown to inhibit human P-gp [13]. Therefore, SAL is likely to affect the pharmacokinetics of ENR through ABC transporters. In contrast, the AUC_0-∞_ of SAL was significantly reduced in the ENR + SAL group compared to the SAL alone group. After continuous administration of ENR for 5 days (ENR (5d) + SAL group), the AUC_0-∞_ and MRT_0-∞_ of SAL increased significantly (Appendix A). This result may have been caused by the induction of CYP3A4 activity by low concentrations of ENR and short treatment of ENR (Figure 2).

To verify the above hypothesis, we took the liver and small intestine of different dosing groups and tested the gene and protein expression of CYP3A4, P-gp and BCRP in tissues. The results of RT-PCR and WB of P-gp and BCRP in liver and intestine showed that ENR and SAL could downregulate the gene and protein levels of P-gp and BCRP in broilers (Figure 5 and Figure 6). However, long-term administration of ENR and SAL upregulates P-gp expression, which has similarities and contradictions with previous reports in the literature. In a study by Sousa et al., SAL was found to induce P-gp in human lung adenocarcinoma cells A549 in a time-dependent manner. A volume 5 μM SAL treatment for 30 min and 1 h had no significant effect on P-gp, but SAL treatment for 72 h significantly induced the expression of P-gp [32]. This phenomenon also exists in ENR. After single oral doses of 20, 40 and 80 mg/kg ENR for 3 h, the liver and kidney of tilapia significantly expressed P-gp [33]. Therefore, according to the concentrations of SAL and ENR in the literature, it is possible that 7.5 mg/kg SAL and 6.0 mg/kg ENR could reduce the expression of P-gp in our study. Additionally, this result is the same as that observed for quercetin in chickens [26]. Meanwhile, SAL was found to inhibit BCRP in a time- and concentration-dependent manner in our study, a result consistent with previous studies [34,35]. The effect of ENR on BCRP has not yet been reported in the literature. Here, it is reported for the first time that ENR induces BCRP in broiler chickens in a time- or concentration-dependent manner, and low concentrations of ENR inhibit BCRP expression. Based on the pharmacokinetic results of ENR and SAL, the significant increase in the plasma concentrations of ENR and CIP after co-administration of SAL and ENR is the result of the inhibition of CYP3A4, P-gp and BCRP expression levels by SAL. While SAL was administered continuously for 5 days, the plasma concentrations of ENR and CIP were slightly decreased compared with the ENR + SAL group corresponding to the significant overexpression of P-gp and BCRP in the liver and small intestine (compared with the single ENR group). The effect of ENR on SAL pharmacokinetics in broilers was also mediated by CYP3A4, P-gp and BCRP.

Subsequently, we explored whether the effects of ENR and SAL on P-gp and BCRP were affected by treatment time and the concentration applied to human rectal adenocarcinoma cell line Caco-2 cells. All drug concentrations in cell assays inhibited P-gp and BCRP gene and protein levels, but the inhibition decreased with time and concentration (Figure 7). This is consistent with the results observed above in broiler liver and small intestine tissues. In conclusion, the inhibition of CYP3A4 activity and the downregulation of CYP3A4 and BCRP expression caused by ENR and SAL under the actual dosage and treatment course, which significantly increased intestinal absorption and hepatic metabolism and clearance of ENR, CIP and SAL, resulted in increased plasma concentrations of ENR, CIP and SAL. This is a serious concern because higher antibiotic concentrations and prolonged elimination times may lead to ENR and SAL poultry food residues, which may indirectly induce the development of resistance.

Nuclear receptors PXR and CAR are ligand-dependent activated transcription factors that are mainly expressed in the liver, small intestine, and colon. Therapeutics, environmental pollutants, and endogenous substances can all act as ligands for PXR and CAR [17,36]. After ligand activation, PXR and CAR translocate protein complexes formed with retinoid X receptors (RXRs) in the cytoplasm to the nucleus, where they bind to the drug-responsive elements of the target gene [37]. PXR has been reported to have important regulatory effects on the expression of CYP3A4, P-gp and BCRP [16,18,19]. CAR, as one of the nuclear receptors, is also involved in the transcriptional regulation of P-gp and BCRP [18,19]. In our study, continuous mixed drink of ENR 5d downregulated the gene and protein expression of CYP3A4 in chickens, while ENR also downregulated the gene and protein expression of P-gp and BCRP. This suggested that ENR may be an antagonist ligand for PXR and/or CAR receptors in chickens, one which can reduce the expression of CYP3A4, P-gp and BCRP in chickens by inhibiting PXR and/or CAR. In addition, SAL significantly downregulated the expression of CYP3A4, P-gp, and BCRP in the case of a single oral administration. After 5 days of mixed feeding, the gene and protein expressions of CYP3A4, P-gp and BCRP in the liver and small intestine of chickens were significantly upregulated by SAL. This may mean that the therapeutic concentration and duration of SAL has a significant impact on its binding to PXR and/or CAR ligands. There are no studies confirming that ENR and SAL are antagonists of nuclear receptors PXR and CAR, but our study confirms that ENR and SAL may have the potential to antagonize PXR and CAR. In DDI, antagonistic nuclear receptor PXR- and CAR-mediated inhibition of metabolic enzymes and transporters reduces the systemic clearance of the drug and can increase the therapeutic concentration of the drug, but carries a certain risk of toxicity. The blocking of PXR and CAR activation during drug treatment can also reduce the expression of drug resistance genes, thereby increasing the body’s drug sensitivity. However, for antibiotic drugs, this effect may also accelerate the development of bacterial resistance. In addition, SAL has been reported to have clear anti-tumor activity after studies this far [38], and the inhibition of PXR can prevent the spread and drug resistance of cancer cells. It is worth studying whether this may also be an anti-tumor mechanism of SAL.

To further elucidate the pharmacokinetic DDI mechanism mediated by ENR and SAL and based on CYP3A4, P-gp and BCRP, computer-assisted molecular docking was performed to understand the binding sites and interaction between ENR and SAL and CYP3A4 (Figure 8), P-gp (Figure 9) and BCRP (Figure 10). The binding of ENR and SAL to CYP3A4 interferes with each other. Due to the sharing of part of the active site, ENR and SAL can homogeneous competitive inhibition of CYP3A4 to catalyze SAL and ENR, respectively. It was shown that ENR and SAL not only downregulate CYP3A4 expression, but also affect each other’s metabolism through active site competition. P-gp and BCRP belong to a family of ATP-binding cassette (ABC) proteins that expel toxic molecules and drugs from cells through ATP-driven conformational changes [39]. The basic structure of both mainly includes two nucleotide-binding domains (NBDs) and two hydrophobic transmembrane domains [40]. Among them, NBD has an ATP-binding site that can provide energy for transporters to deliver drugs [41]. Through molecular docking, it was found that ENR and SAL have a certain binding affinity with NBD of P-gp and BCRP, and both share a common binding site with ATP. This competition for binding sites may affect the efficient binding of P-gp and BCRP to ATP, resulting in reduced transporter energy supply, thereby inhibiting the drug efflux function of P-gp and BCRP.

In conclusion, our results suggest that ENR and SAL regulate the expression of CYP3A4, P-gp and BCRP in Caco-2 cells as well as in the liver and small intestine of broilers, and may structurally inhibit the actions of CYP3A4 and transporters through a steric-like effect. Thus, there is a pharmacokinetic DDI based on CYP3A4, P-gp, and BCRP occurs when ENR and SAL, two drugs commonly found in broiler farming, are co-existed in broilers. This study is the first to report the pharmacokinetic DDI between ENR and SAL, and the first to explore the role of metabolic enzymes and hepatic and gut transporters in inducing the two drug DDIs.

The widespread use of ENR and anticoccidial products has led to severe drug resistance, which is a serious challenge for the entire poultry industry [42]. Additionally, our research shows that, under the actual concentration and course of treatment, ENR and SAL produce DDI and may lead to drug residual and resistance. Additionally, longer coexistence may also lead to lower drug bioavailability and lower efficacy. Therefore, there is an urgent need to further evaluate the long-term administration of SAL or ENR on the residues of ENR or SAL in broilers. The means of rationally using antibiotics has always been a subject of great research and discussion. The results of this study have important implications for the rational drug use in poultry breeding. That is, the fact ENR and SAL are not suitable for broilers to take together, and it is necessary to further consider whether SAL is suitable to be added to the feed. In addition, the DDI of SAL, a feed additive, and other antibiotics which were mediated by CYP3A4 metabolism, P-gp and BCRP efflux also deserve further study.

## Figures and Tables

**Figure 1 antibiotics-12-00403-f001:**
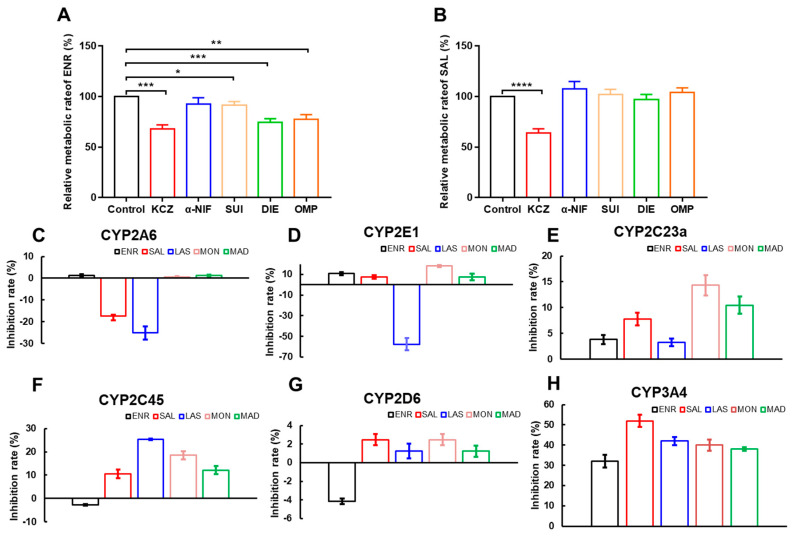
The effects of several positive inhibitors on the relative metabolic rate of ENR (**A**) and SAL (**B**). Effects of SAL, LAS, MAD, MON and ENR on broiler CYP2A6 (**C**), CYP2E1 (**D**), CYP2C23a (**E**), CYP2C45 (**F**), CYP2D6 (**G**) and CYP3A4 (**H**) activity. Under the 100 µmoL/L, the five drugs all showed the inhibition activity to CYP3A4. KCZ: ketoconazole; α-NIF: alpha-naphthoflavone; SUI: sulfapyrazole; DIE: sodium diethyldithiocarbamate; OMP: omeprazole. SAL, salinomycin; LAS, lasalocid; MAD, maduramycin; MON, monensin; ENR, enrofloxacin. * means *p* < 0.05; ** means *p* < 0.01; *** means *p* < 0.005; **** means *p* < 0.001.

**Figure 2 antibiotics-12-00403-f002:**
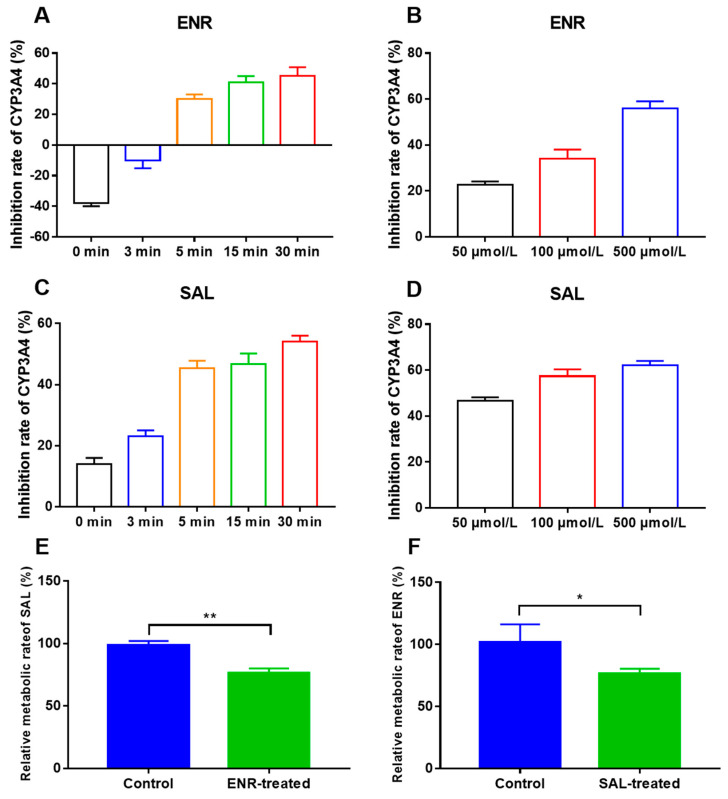
The effects of different pre-incubation times and drug concentrations on the metabolic activity of CYP3A4. (**A**,**B**) are the inhibition rates of CYP450 enzyme activity with different concentrations of ENR and 100 μmol/L ENR pre-incubated for 0, 3, 5, 15, and 30 min, respectively. (**C**,**D**) are the inhibition rates of CYP450 enzyme activity with different concentrations of SAL and 100 μmol/L ENR pre-incubated for different times, respectively. (**E**) is the relative metabolism rate of SAL in the presence of ENR; (**F**) is the relative metabolism rate of ENR in the presence of SAL. * means *p* < 0.05; ** means *p* < 0.01.

**Figure 3 antibiotics-12-00403-f003:**
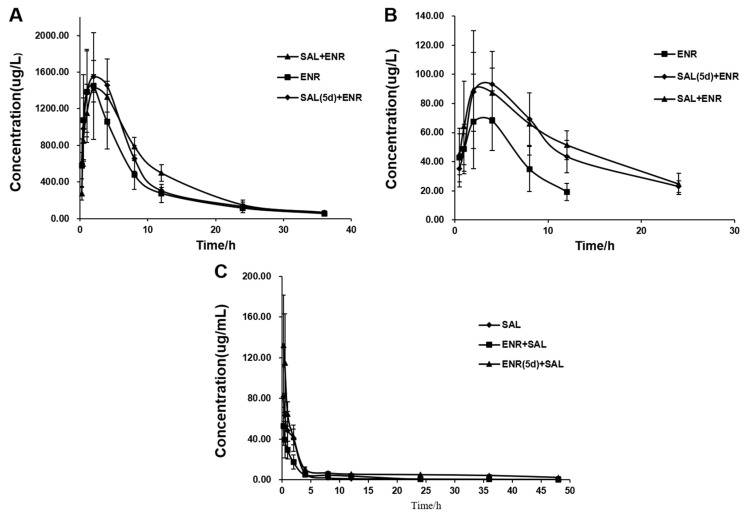
Drug concentration–time curves of ENR (**A**), CIP (**B**) and SAL (**C**) in broiler plasma after single or combined administration (*n* = 6).

**Figure 4 antibiotics-12-00403-f004:**
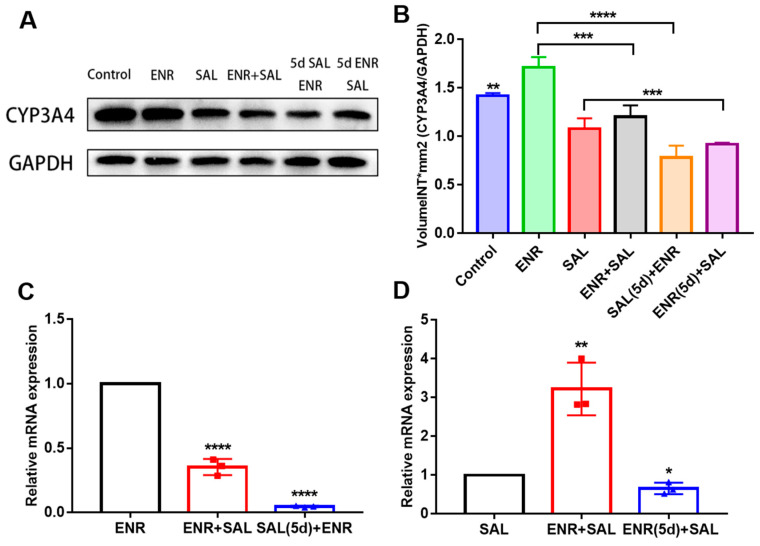
Effects of ENR and SAL on CYP3A4 gene expression levels in liver. (**A**) is the immunoblotting image and (**B**) is the gray value of the immunoblot band; (**C**) is the effects of single ENR as control group, single combined administration group (ENR + SAL) and continuous 5d SAL group (SAL (5d) + ENR) on the mRNA expression level of CYP3A4. (**D**) is the effects of single SAL as control group, single combined administration group (ENR + SAL) and continuous 5d ENR group (ENR (5d) + SAL) on the mRNA expression level of CYP3A4. ** on the control group indicates a significant difference between the other groups and the control group, *p* < 0.01 (**B**). * means *p* < 0.05; ** means *p* < 0.01; *** means *p* < 0.005; **** means *p* < 0.001.

**Figure 5 antibiotics-12-00403-f005:**
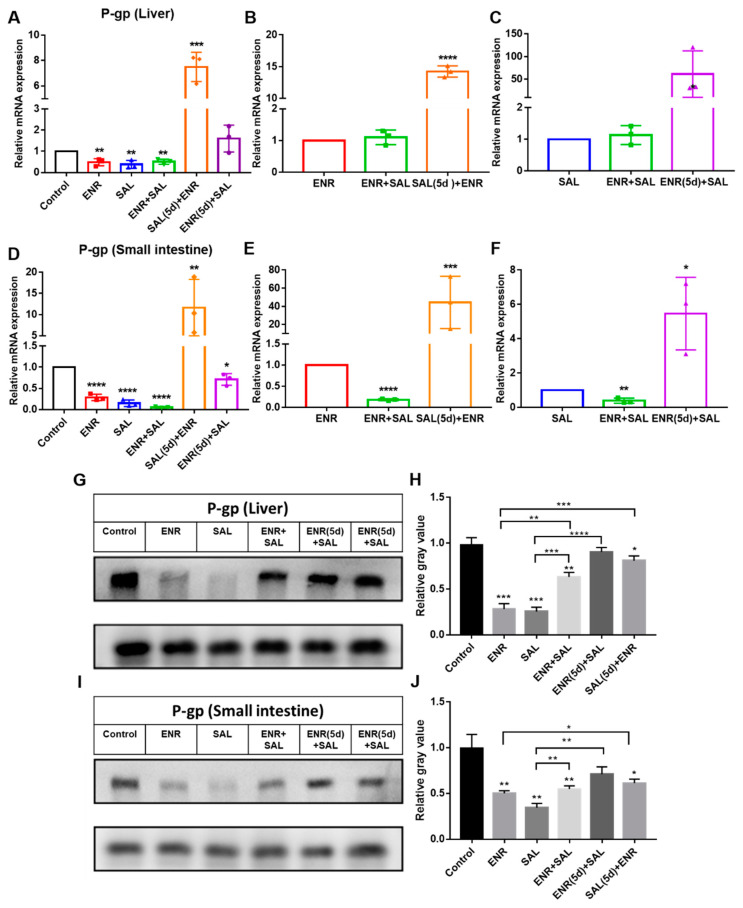
Effects of ENR and SAL on P-gp gene expression levels in liver and small intestine. (**A**–**C**) are expression in liver, (**D**–**F**) are expression in small intestine. (**B**,**E**) are the effects of single ENR as control group, single combined administration group (ENR + SAL) and continuous 5d SAL group (SAL (5d) + ENR) on the expression level of P-gp. (**G**,**I**) are protein expression in liver and small intestine, respectively. (**H**,**J**) are gray value levels of figure (**G**) and (**I**), respectively. (**C**,**F**) are the effects of single SAL as control group, single combined administration group (ENR + SAL) and continuous 5d ENR group (ENR (5d) + SAL) on the expression level of P-gp. * on the error bar of each group indicates the difference from the control group. * means *p* < 0.05; ** means *p* < 0.01; *** means *p* < 0.005; **** means *p* < 0.001.

**Figure 6 antibiotics-12-00403-f006:**
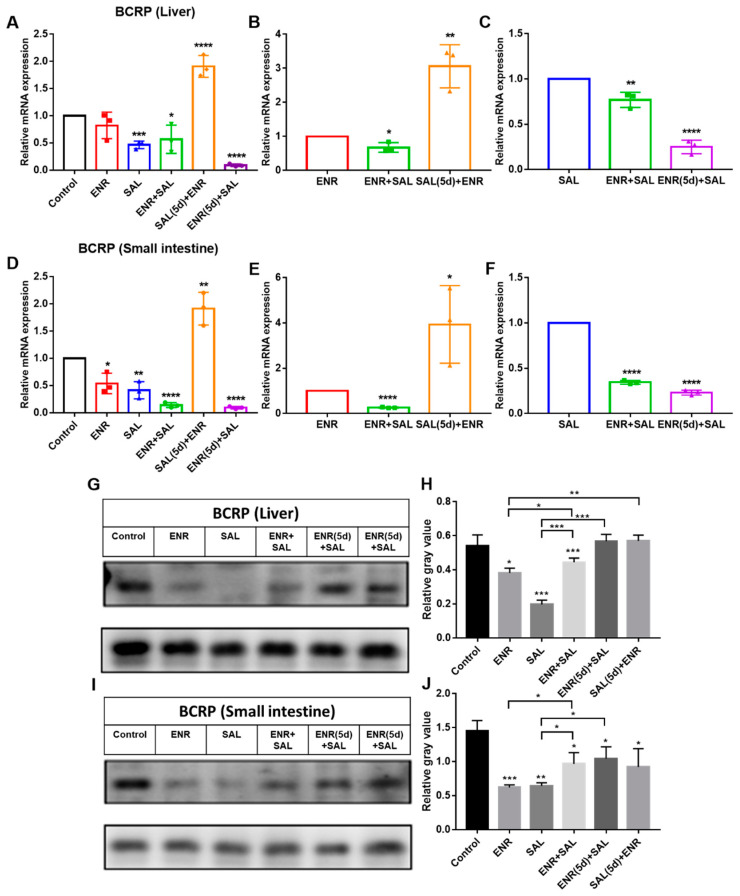
Effects of ENR and SAL on BCRP gene expression levels in liver and small intestine. (**A**–**C**) are expression in liver, (**D**–**F**) are expression in small intestine. (**B**,**E**) are the effects of single ENR as control group, single combined administration group (ENR + SAL) and continuous 5d SAL group (SAL (5d) + ENR) on the expression level of BCRP. (**G**,**I**) are protein expression in liver and small intestine, respectively. (**H**,**J**) are gray value levels of figure (**G**) and (**I**), respectively. (**C**,**F**) are the effects of single SAL as control group, single combined administration group (ENR + SAL) and continuous 5d ENR group (ENR (5d) + SAL) on the expression level of BCRP. * on the error bar of each group indicates the difference from the control group. * means *p* < 0.05; ** means *p* < 0.01; *** means *p* < 0.005; **** means *p* < 0.001.

**Figure 7 antibiotics-12-00403-f007:**
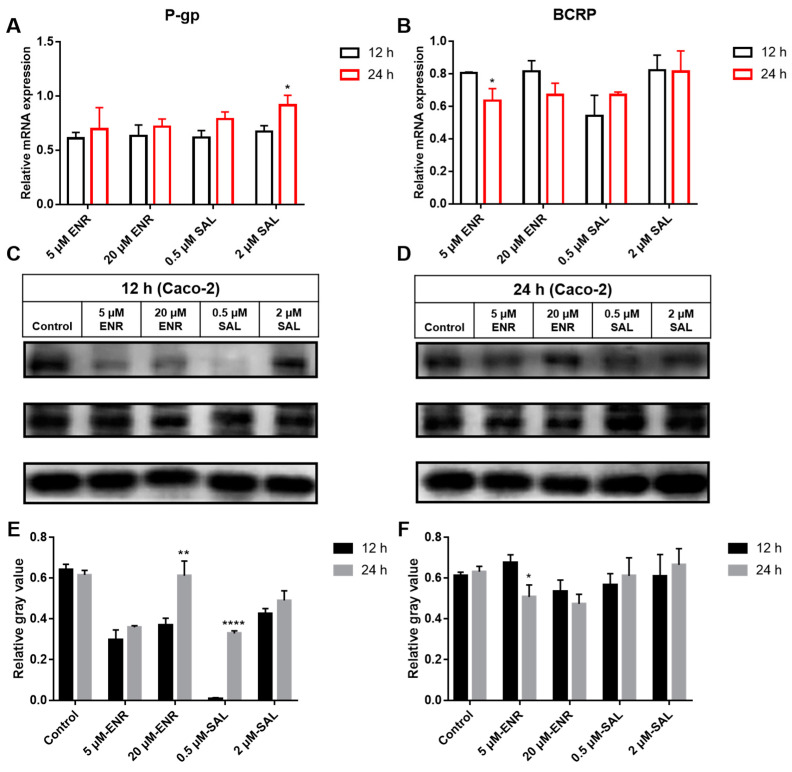
Effects of ENR and SAL on gene and protein expression of P-gp and BCRP in Caco-2 cells. (**A**,**B**) are the mRNA expression levels of P-gp and BCRP, respectively. (**C**,**D**) are the protein expression levels of P-gp and BCRP, respectively. (**E**,**F**) are gray value levels of P-gp (**C**) and BCRP (**D**), respectively. * indicates the difference compared with the same concentration of drug incubated for 12 h. * means *p* < 0.05; ** means *p* < 0.01; **** means *p* < 0.001.

**Figure 8 antibiotics-12-00403-f008:**
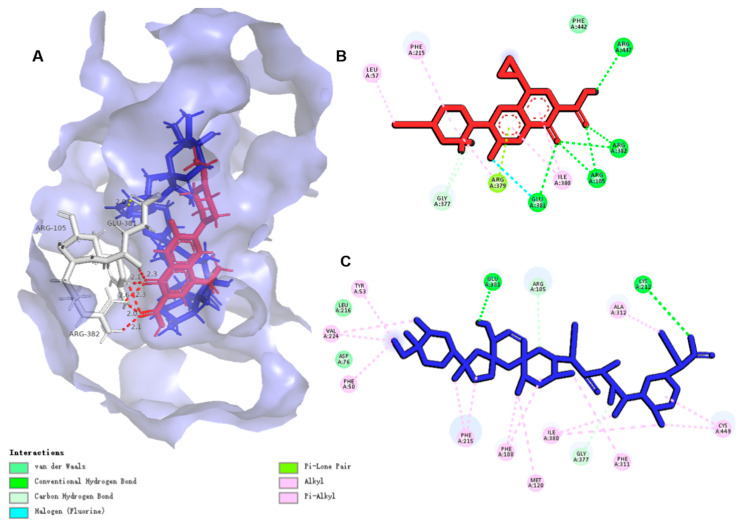
Molecule docking simulations of compounds with CYP3A4 (PDB ID: 6MA7). (**A**) ENR and SAL in the active area of CYP3A4 (PDB ID: 6MA7). (**B**,**C**) are 2D maps of the interaction of ENR and SAL with CYP3A4, respectively. The green dashed lines indicate hydrogen bonding forces between compounds and amino acid residues.

**Figure 9 antibiotics-12-00403-f009:**
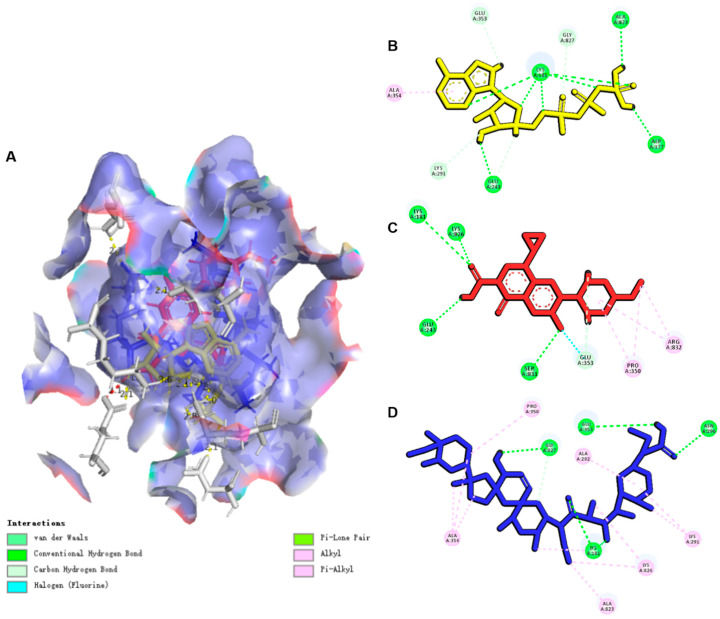
Molecule docking simulations of compounds with P-gp (PDB ID: 6C0V). (**A**) ENR (Red), SAL (Blue) and ATP (Yellow) in the active area of P-gp (PDB ID: 6C0V). (**B**–**D**) are 2D maps of the interaction of ENR, SAL and ATP with P-gp, respectively. The green dashed lines indicate hydrogen bonding forces between compounds and amino acid residues.

**Figure 10 antibiotics-12-00403-f010:**
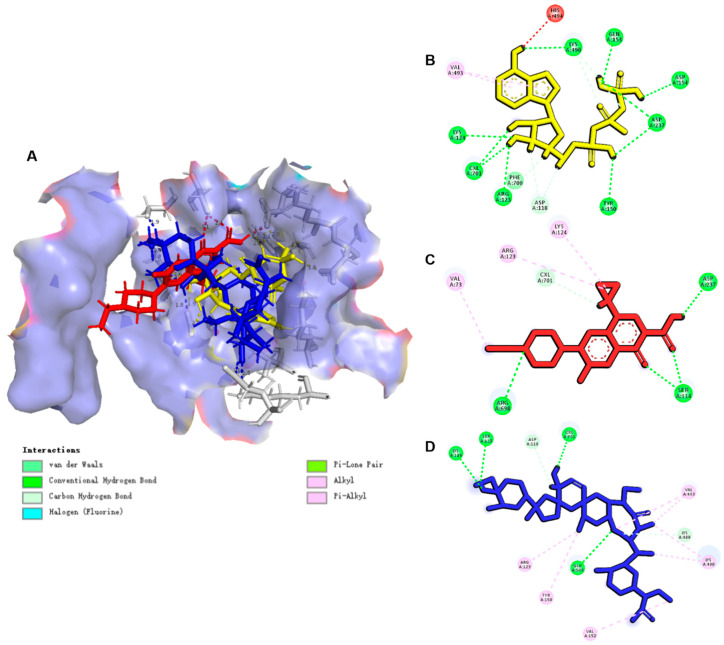
Molecule docking simulations of compounds with BCRP (PDB ID: 6ETI). (**A**) ENR (Red), SAL (Blue) and ATP (Yellow) in the active area of BCRP (PDB ID: 6ETI). (**B**–**D**) are 2D maps of the interaction of ENR, SAL and ATP with BCRP, respectively. The green dashed lines indicate hydrogen bonding forces between compounds and amino acid residues.

## Data Availability

Not applicable.

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
