# Peer review of "ABC Transporters and CYP3A4 Mediate Drug Interactions between Enrofloxacin and Salinomycin Leading to Increased Risk of Drug Residues and Resistance"

_antibiotics, 2023, doi:10.3390/antibiotics12020403_

Round 1

Reviewer 1 Report

Journal- Antibiotics

Title- "ABC transporters and CYP3A4 mediate drug interactions between enrofloxacin and salinomycin leading to increased risk of drug residues and resistance"

Reviewers' comments

Reviewers' comments

1.   What is the criterion for selecting the doses of ENR and SAL for the in-vitro and in-vivo experiments?

2.   The methodology of RT-PCR and Western blot is not adequately explained. The authors must explain it properly and mention the treatment concentration and time.

3.   What are the passage numbers, culture conditions, and incubation time of Caco2 cells? Authors need to mention it in detail.

4.   The 500 μM concentration of ENR is too high, and such concentration may induce stress. Authors need to avoid such high concentrations in cell viability assay.

5.   The doubling time of Caco-2 cells is 62 h. In 12 h, these cells do not form a proper colony (possible in the case of a fast-growing cell line like MDR1-MDCK), so why did the author select the 12 h treatment for mRNA expression? what was the cell count during the experiment?  Why did not use any positive control?

6.   Authors need to give the NCBI gene number of various genes shown in Table S1

7.   Phytochemicals (from food and dietary supplements) and xenobiotics (pharmaceutical drugs and toxins) acts as the ligand for nuclear receptors (PXR, AhR and CAR) and activates their downstream genes (CYPs and drug transporters). Simultaneously, they also antagonise translationally matured CYPs and P-gp and attenuate their catalytic activity. Because this article is on CYP3A4 and P-gp transporter, authors can use this concept in an introduction and discussion section. For this, authors should use the recent articles as https://doi.org/10.1016/j.phymed.2020.153416; https://doi.org/10.1002/fft2.110 and  https://doi.org/10.3389/fendo.2022.959902.

8.   Are there any other interactions apart from hydrogen bonding? If so, the authors should mention it. What was the hydrogen bond distance?

9.   The methodology and results are written very poorly. Besides, enormous grammatical and typological errors are found throughout the manuscript.

Decision: Major Revision

Author Response

  1. What is the criterion for selecting the doses of ENR and SAL for the in-vitro and in-vivo experiments?

Answer: Thank you for your advice. The dose and timing of in vitro pharmacokinetic administration of ENR and SAL is the recommended dose of enrofloxacin and salinomycin in veterinary clinics. The recommended dose and usage of ENR in veterinary clinical practice is: when administered orally, poultry is 5~7.5 mg/kg, twice a day, continuous use for 3~5 d; When ENR is mixed drinking, poultry is 50~75 mg/mL, and it is used for 3~5 d. The recommended dosage and usage of SAL in veterinary clinics is: mixed feeding, chicken is 60 g / 1000 kg feed. Therefore, a single oral dose of ENR was determined to be 7.5 mg/mL. The single oral administration of SAL was 6.0 mg/mL, which was calculated based on poultry weight and feed intake. Multiple consecutive doses were consistent with the recommended clinical dosing, and the ENR pretreatment group was mixed drinking with 75 mg/L water at a continuous dose for 5 d. The SAL pretreatment group was mixed feeding at a dose of 60 mg/kg feed for 5 d. The concentrations of ENR and SAL for the in-vitro probe test were pre-experiments based on a series of concentrations reported in the available literature, and then drug concentrations of 50, 100, and 500 μM were determined. ENR and SAL were used as substrates in vitro chemical inhibitor test, and the incubation system was optimized from three aspects: protein concentration, reaction time and substrate concentration at a fixed temperature, and finally determined that ENR and SAL of 100 μM were the optimal concentrations.

  1. The methodology of RT-PCR and Western blot is not adequately explained. The authors must explain it properly and mention the treatment concentration and time

Answer: We have revised the methodology of RT-PCR and Western blot (2.9. RT-PCR and Western-blot, line 218-242).

  1. What are the passage numbers, culture conditions, and incubation time of Caco2 cells? Authors need to mention it in detail.

Answer: Thank you for your advice. The number of passages when the cells were purchased back is 5, and the passage of Caco-2 cells at the time of stimulation by ENR and SAL was approximately 12-14 passages. Caco-2 cells were grown using DMEM (Gibco, China) containing 10% fetal bovine serum (FBS, PAN, Germany) and 1% 100 U/mL penicillin/streptomycin (Gibco, China) and cultured in a 37 °C constant temperature incubator containing 5% CO2. Cells could adhere to the wall about 6-7 h after passage, and grow to 80% in 24-36 hours. The details are set out in the manuscript methodology (line 132-138).

  1. The 500 μM concentration of ENR is too high, and such concentration may induce stress. Authors need to avoid such high concentrations in cell viability assay.

Answer: Thank you for your advice. When we performed the ENR in Caco-2 cell viability assay, we determined that 500 μM was the last in a series of concentrations in the cell viability assay. On the one hand, it is necessary to be consistent with the in vitro incubation concentration, and on the other hand, this concentration has been reported in the previous literature in cell viability tests. Finally, according to the cell viability assay, two concentrations, 5 and 20 μM, were selected for subsequent RT-qPCR assay and WB assay. Thank you very much for your suggestions, we will definitely consider your suggestions in subsequent studies.

  1. The doubling time of Caco-2 cells is 62 h. In 12 h, these cells do not form a proper colony (possible in the case of a fast-growing cell line like MDR1-MDCK), so why did the author select the 12 h treatment for mRNA expression? what was the cell count during the experiment?  Why did not use any positive control?

Answer: Thank you for your advice. When we culture Caco-2 cells, passage at 1:2, Caco-2 cells can grow to 80% of the T25 cell culture flask in 24-36 h, and the cell density is about 5×106. We plate Caco-2 cells and allow them to grow to 70% (about 12 h, colonies of cells have formed, similar to islands) for drug stimulation, and then measure the gene expression of P-gp at 12 h and 24 h of drug incubation. The absence of a positive control drug is a deficiency of this study, and we can supplement the trial if needed.

  1. Authors need to give the NCBI gene number of various genes shown in Table S1.

Answer: Thank you for your advice. We have supplemented the NCBI gene number of the detected gene in Table S1 and Table S2.

Table S1. Real-time PCR primers for Caco-2 cells

Gene

Forward primer (F)

Reverse primer (R)

P-gp

(NM_001348945.2)

CTAATGGCTTTCCTTCTGATGC

TAAGCTGATAGACGTCAGACAC

BCRP

(NM_004827.3)

GCAGCAGGTCAGAGTGTGGTTTC

ACTGAAGCCATGACAGCCAAGATG

GAPDH

(NM_002046.7)

CATGTTGCAACCGGGAAGGA

CAGGAGCGCAGGGTTAGTC

Table S2. Real-time PCR primers for broiler liver and small intestine

Gene

Forward primer (F)

Reverse primer (R)

CYP3A4

(NM_001329508.2)

GTGGACTTCCTGCAGCTGAT

CCTTCTCCCTGGCAGACTTG

P-gp

(NM_204894.2)

GCTGACTGTGTAGGGACTCA

GGTCCAGTTGGCCTGCAAAT

BCRP

(NM_001328490.2)

CCGCTTGTCCACCAGTTACTTCAG

TTGCCATGTTAGTAGGTGCGATTCC

GAPDH

(NM_204305.2)

GCAACCGTGTTGTGGACTTG

CTCCAACAAAGGGTCCTGCT

  1. Phytochemicals (from food and dietary supplements) and xenobiotics (pharmaceutical drugs and toxins) acts as the ligand for nuclear receptors (PXR, AhR and CAR) and activates their downstream genes (CYPs and drug transporters). Simultaneously, they also antagonise translationally matured CYPs and P-gp and attenuate their catalytic activity. Because this article is on CYP3A4 and P-gp transporter, authors can use this concept in an introduction and discussion section. For this, authors should use the recent articles as https://doi.org/10.1016/j.phymed.2020.153416; https://doi.org/10.1002/fft2.110 and  https://doi.org/10.3389/fendo.2022.959902.

Answer: Thank you for your advice. We discussed the link between nuclear receptors and metabolic enzymes and transporters in the introduction and discussion sections (line 97-110; line 605-634).

  1. Are there any other interactions apart from hydrogen bonding? If so, the authors should mention it. What was the hydrogen bond distance?

Answer: Thank you for your advice. We have added other interactions and hydrogen bond distances (line 683-figure 8, line 503-figure 9 and line 508-figure 10).

  1. The methodology and results are written very poorly. Besides, enormous grammatical and typological errors are found throughout the manuscript.

Answer: Thank you for your advice. We have made modifications to the methodology and results. We hope the revised paper will be more clear and accurate on expressions.

Reviewer 2 Report

I have several comments that should be clarified and explained:

1. Please present more key findings in the abstract

2. The abbreviations should be defined just the first time they are mentioned in the text and used as such throughout it.

3. Please improve the discussion section and comparison of the results with the literature data. Overall, the discussion section has to be improved. 

4. The importance of this study and the practical applications of the findings has to be expressed more.

5. The English language and style must be improved, it is difficult to follow and understand some paragraphs. 

Author Response

  1.  Please present more key findings in the abstract.

Answer: Thank you for your advice. We have made modifications to the abstract (line 20-30).

  1. The abbreviations should be defined just the first time they are mentioned in the text and used as such throughout it.

Answer: Thank you for your advice. We have read through the full text and carefully checked to ensure that abbreviations are defined when they are first mentioned in the text.

  1. Please improve the discussion section and comparison of the results with the literature data. Overall, the discussion section has to be improved. 

Answer: Thank you for your advice. We revised the discussion and also added comparisons of our results with the literature data (line 570-581 are the comparison of the results in this study and literature).

  1. The importance of this study and the practical applications of the findings has to be expressed more.

Answer: Thank you for your advice. We have added a paragraph after the conclusion to explain the importance and practical application of this research (line 652-672).

  1. The English language and style must be improved, it is difficult to follow and understand some paragraphs.

Answer: Thanks very much for your comments, which are very helpful for us to improve the manuscript, and our language should be improved. After carefully check, we found many grammar and sentence errors, and have modified the manuscript accordingly. We hope the revised paper will be more clear and accurate on expressions.

Round 2

Reviewer 1 Report

Journal- Antibiotics

Title- "ABC transporters and CYP3A4 mediate drug interactions between enrofloxacin and salinomycin leading to increased risk of drug residues and resistance"

Reviewers' comments

Dear Editor,

Authors have fulfilled all the queries/comments as it was asked previously. Now the manuscript is well written. I believe that it is a nice piece of work for being published in the Antibiotics. Finally, I recommend that the paper should be accepted for the publication in the present form.

Decision- Accept

Reviewer 2 Report

Dear authors,

I have no other comments. 

Good luck!